# A New Creep–Fatigue Interaction Model for Predicting Deformation of Coarse-Grained Soil

**DOI:** 10.3390/ma15113904

**Published:** 2022-05-30

**Authors:** Jie Zhang, Qiuhua Rao, Wei Yi

**Affiliations:** School of Civil Engineering, Central South University, Changsha 410075, China; jinxi625@live.cn (J.Z.); yi.wei@csu.edu.cn (W.Y.)

**Keywords:** creep–fatigue interaction, coarse-grained soil, rheological mechanic, creep–fatigue interaction test, nonlinear regression

## Abstract

Studying the creep–fatigue interaction of the coarse-grained soil (CGS) is very important for safety assessment and disaster prevention in subgrade engineering. Current research work is mainly focused on single creep or fatigue deformation. In this paper, a new creep–fatigue interaction model is established to predict the creep–fatigue interaction deformation of different gradation CGS based on the rheological mechanics and the interactive relationship between creep and fatigue complex compliance method. Triaxial creep–fatigue interaction tests of different gradations CGS under different average stresses and frequencies were conducted to verify the new creep–fatigue interaction model. Research results show that for the creep–fatigue and fatigue–creep interaction, the fatigue deformation is always larger than the creep deformation under the same stress level. For the creep–fatigue multi-interaction, the second creep and fatigue deformation are always smaller than the first creep and fatigue deformation. The results of the triaxial creep–fatigue interaction tests verify the validity of this new model.

## 1. Introduction

Coarse-grained soil (CGS in short) is a mixture of pebble, sand and clay, etc. [1,2], and is commonly used as a subgrade material in highway and railway engineering [3,4,5]. It is generally subjected to static or repeated dynamic load caused by transportation tools (vehicles and trains) [6,7]. Since pebble and sand usually show elasticity while clay has viscoelasticity, the CGS is considered to be of rheological properties [8]. Under long-term static–dynamic (i.e., creep–fatigue) interaction loading, the CGS may cause uneven settlement and potential safety hazards for the highway and railway [9].

Currently, there are two main creep–fatigue loading forms: the simultaneous superposition of creep and fatigue loads [10,11] and alternative creep and fatigue loads [12,13], among which the latter is a common case for the CGS of subgrade in highway and railway engineering since the vehicles and trains usually have an alternative state of stopping or moving (i.e., successive creep–fatigue loads). Therefore, it is of great significance to study the successive creep–fatigue interacting models and mechanisms for predicting the long-term deformation of CGS. Niya Dong [14] proposed a newly modified multi-sequenced repeated load test, measured the strain, resilient modulus, and creep compliance, proposed new factors to evaluate the viscoelasticity of different asphalt mixtures. V. P. Golub [15] proved that there exists a definite correlation between the processes of static and cyclic creep and proposed a relationship of creep–fatigue interaction. Yunliang Li [16] analyzed the cement emulsified asphalt composite binder evolution process of fatigue damage by fatigue test, and with the increase in fatigue damage, instantaneous elastic deformation and creep rate increase in steady creep stage during creep process. Linjian Ma [13] conducted the creep–fatigue interaction test of rock salt by trapezoidal wave, and the results showed that under combined creep and fatigue, the cycling life of the rock salt is remarkably lower compared with that observed in pure fatigue or creep tests and the life of creep–fatigue interaction can be alternatively predicted without rupturing the salt specimens. Hongfu Liu [11] established a fatigue–creep damage interaction model considering the interaction of fatigue and creep damage based on the principle of semi-sine fatigue stress decomposition. The results indicated that the damage of the asphalt mixture under the action of the semi-sine cyclic load was not a simple linear accumulation of fatigue damage and creep damage, but an overall result of the interaction of fatigue damage, creep damage, and fatigue–creep damage. Qian Li [17] researched and indicated that cyclic loading has a significant influence on the creep of high-strength high-performance concrete, especially during the early stage of creep development. Additionally, the modified creep model can be adopted to predict the long-term behaviors of concrete bridges, which are always subjected to cyclic traffic loading. Bara Wasfi Al-Mistarehi [18] considered that the accumulated strain which increased with time at different levels of temperature for all types of fillers was obtained from the creep test and fatigue test and between logarithm strain and logarithm number of cycles. As a result, the waste toner as filler was the best filler. Rafiqul Tarefder [19] showed that the increase in freeze–thaw cycles in asphalt concrete decreased the indirect tensile strength and tensile strength ratio of asphalt. Xiaozhao Li [20] proposed an analytical solution by coupling the confined cyclic static loading and unloading path, the Hooke–Kelvin viscoelastic model and the formulated micro–macro model which explains the total time-dependent visco–elastic–plastic deformation caused by the microcracks variable during cyclic static compressive failure. Vitaliy M. Kindrachuk [21] developed a fatigue model for normal strength concrete under compressive loading conditions based on continuum damage mechanics which incorporates creep and fatigue phenomena in a single framework and postulates damage to be driven by inelastic deformations. Yu Su [5] developed a fatigue model of railway soil that considered the coarse grain content and the number of loading cycles. Abdulgazi Gedik [22] studied the fatigue and dynamic creep modulus of asphalt pavement with recycling fluorescent lamps and Yasser M [23] studied the dynamic rheological characteristics of asphalt pavement with two different recycled plastic wastes. Estelle Delfosse Ribay [24] presented the creep and fatigue behaviors of grouted sand to analyze the strain rate, creep slope or fatigue slope, and the creep limit strength or fatigue limit strength. Alireza Ameli [25] analyzed the resilient modulus, indirect tensile strength and dynamic creep of asphalt binders and mixtures. Very few studies are available on the creep–fatigue interaction of the CGS.

In this paper, two new interaction parameters are proposed to establish the new creep–fatigue interaction model based on rheology mechanics and the relationship between the creep and fatigue complex compliance. The interaction parameters were determined by the triaxial creep and fatigue tests, and the new interaction model was verified by the predicted and tested results of the triaxial creep–fatigue interaction tests.

## 2. Establishment of New Creep–Fatigue Interaction Model

### 2.1. Definition of the New Creep–Fatigue Interaction Factor

Figure 1 shows the creep–fatigue interaction process of the CGS under triaxial compression (*σ*_1_
*> σ*_3_, *σ*_1_ and *σ*_3_ are axial and confining stresses), where the interaction order can be the creep first followed by fatigue (creep–fatigue interaction, C-F for short), or the fatigue first followed by creep (fatigue–creep interaction, F–C in short). For each creep–fatigue interaction process, the creep stress (*σ*_1_
*= σ_c_*, *σ_c_* is the creep stress, *σ_c_* − *σ*_3_ is the deviatoric stress of creep) is applied to the CGS for a period of time (*t* = *t*_1_) and unloaded to be *σ*_1_ = *σ*_3_*,* and then the fatigue stress in terms of semi-sine (*σ*_1_ = *σ_f_* = 2*σ_c_*|sin*ωt*|, *ω* is circular frequency, *ω* = 2*πf*, *f* is fatigue loading frequency) is applied to the CGS for a period of time (*t* = *t*_2_ − *t*_1_). Let the creep deviatoric stress (*σ_c_* − *σ*_3_) be equal to the average deviatoric stress of the fatigue cycle (*σ_a_ = σ_c_* − *σ*_3_, *σ_a_* is the average deviatoric stress of fatigue, *σ_a_* = (*σ_f_* − *σ*_3_)/2) since in order to establish a relationship under the same stress factor.

In order to study the creep–fatigue interaction for the CGS deformation, it is necessary to seek a creep–fatigue interaction parameter to link up the two loading processes.

In rheologic mechanics, there exists a relationship of Fourier equivalent transformation between the creep compliance *J*(*t*) and fatigue compliance *J*(*ω*), where ℱ is Fourier transformation and ℱ^−1^ is inverse Fourier transformation.
(1)$$\begin{array}{l} J(\omega )=ℱ[J(t)]\\ J(t)={ℱ}^{-1}[J(\omega )] \end{array}$$

Since test results show that the CGS has larger fatigue deformation than creep deformation under *σ_a_* = *σ_c_* [26], *J*(*t*) and *J*(*ω*) cannot be equivalently transformed by Equation (1). For modifying Equation (1) to obtain the actual equivalent transform relationship between *J*(*t*) and *J*(*ω*) of the CGS, *J*(*ω*) is divided into a real part *J_r_*(*ω*) and imaginary part *J_i_*(*ω*), and *J*(*t*) can also be in terms of its real part *J_r_*(*t*) and imaginary part *J_i_*(*t*) by Fourier transform.
(2)$$\begin{array}{l} J_{r}(t)=\omega \int_{0}^{\infty }J(t)\sin\omega tdt\\ J_{i}(t)=\omega \int_{0}^{\infty }J(t)\cos\omega tdt \end{array}$$

Considering that both the creep and fatigue compliances have real and imaginary parts, define two new creep–fatigue interaction factors *k* and *g*, define the matrices *A*_1×2_ = (*J_r_*(*t*) *J_i_*(*t*)), *B*_1×2_ = (*J_r_*(*ω*) *J_i_*(*ω*)) and *M*_2×2_ =$\;\left(\begin{array}{cc} k & 0\\ 0 & g \end{array}\right)$, and then establish the relationship between the real and imaginary parts of the creep (*J_r_*(*t*), *J_i_*(*t*)) and fatigue (*J_r_*(*ω*), *J_i_*(*ω*)) compliances, i.e., the new creep–fatigue interaction model:(3)$$\begin{array}{l} \left(\begin{array}{cc} J_{r}(t) & J_{i}(t) \end{array}\right)\left(\begin{array}{cc} k & 0\\ 0 & g \end{array}\right)=\left(\begin{array}{cc} J_{r}(\omega ) & J_{i}(\omega ) \end{array}\right)\\ \left(\begin{array}{cc} J_{r}(\omega ) & J_{i}(\omega ) \end{array}\right){\left(\begin{array}{cc} k & 0\\ 0 & g \end{array}\right)}^{-1}=\left(\begin{array}{cc} J_{r}(t) & J_{i}(t) \end{array}\right) \end{array}$$

For simplifying the expression, that is Equation (5):(4)$$\begin{array}{l} AM=B\\ BM^{-1}=A \end{array}$$

Since *J_r_*(*t*), *J_i_*(*t*) and *J_r_*(*ω*), *J_i_*(*ω*)) are obtained by the following creep and fatigue tests, respectively, *k* and *g* can be calculated by Equation (4).

### 2.2. Determination of Creep–Fatigue Interaction Factor

In order to determine the creep–fatigue interaction parameters *k* and *g*: (1) conduct the triaxial creep tests (creep stress *σ_c_*) and the fatigue tests (fatigue average stress *σ_a_*) under the same average stress condition (*σ_c_* − *σ*_3_ = *σ_a_*) to determine creep compliance and fatigue compliance; (2) calculate the creep and fatigue real and imaginary parts of compliance; and (3) compare the real parts and imaginary parts of the creep and fatigue compliance to determine the *k* and *g*.

#### 2.2.1. Scheme of Triaxial Creep and Fatigue Tests

According to the Code for the Design on Subgrade of Railway in China (TB10001-2005), three different gradations of CGS were prepared by different mass ratios of pebble (*γ*) (with high elastic modulus), sand and clay (with rheological properties), as listed in Table 1. Figure 2 and Figure 3 showed their grain–size distribution curves and water content (*w*)–dry density (*ρ_d_*) curves for three different gradations of CGS, respectively. The CGS specimens are cylinders of Φ300 mm × 600 mm and prepared by the layered compacting method (Figure 4).

Triaxial compressive creep and fatigue tests were conducted on the TAJ-2000 triaxial testing system (Figure 5). Table 2 lists their loading conditions for the three different gradations of CGS specimens, where the confining pressure *σ*_3_ was 0.2 MPa (*σ*_3_ < *σ*_1_). Let the average stress of fatigue test *σ_a_ = σ_c_* − *σ*_3_. There were a total of 9 creep tests (3 specimen × 3 stress) and 27 fatigue tests (3 specimen × 3 stress × 3 frequency). For each test, three identical specimens were prepared and there were 108 specimens in total. During the creep tests, the strain was recorded every 2 min to obtain the creep curves. During the fatigue tests, the strain and stress were recorded 10 times in one cycle.

#### 2.2.2. Triaxial Creep Test and Analyses

(1)Creep parameters under different stress

In this study, the Burgers creep model (Figure 6, Equation (5)) was adopted to describe the creep phenomenon of the different gradations of CGS and the creep compliance *J*(*t*) can be obtained by fitting the tested creep curves:(5)$$J(t)=\frac{1}{E_{1}}+\frac{t}{\eta_{1}}+\frac{1-e^{-\frac{tE_{2}}{\eta_{2}}}}{E_{2}}$$
where *E*_1_, *E*_2_, *η*_1_, *η*_2_ are creep parameters of the Burgers model.

Figure 7 shows the tested creep strain data (*ε_c_*) under different time (*t*) curves for different constant stress (*σ_c_* = 0.3 Mpa, 0.4 Mpa, 0.5 Mpa) for the CGS specimens of three different gradations, as well as their fitted curves by the Burgers creep model (Equation (14)). Table 3 lists the fitted Burgers model parameters with high fitting precision (*R*^2^ > 0.995). It is seen that all the creep curves have three stages: the instantaneous deformation stage (*t* = 0), attenuation deformation stage (with decreasing slope), and stable creep deformation stage (with a constant slope). Obviously, for the same gradation of the CGS specimen, *ε_c_* is increased with the increase in *σ_c_*. While for the different gradations of the CGS specimen, *ε_c_* is increased with the decrease in pebble content *γ* since a smaller *γ* means more clay content with stronger rheological properties.

(2)Unified creep parameters for different creep stresses and gradations

It was found in Table 3 that the Burgers creep parameters depend not only on the creep stress (*σ_c_*) or the deviatoric stress (*σ* = *σ_c_* − *σ*_3_), but also on the gradations (mass ratios *γ*, water content *w*). In order to obtain a unified creep parameter relationship (related to *σ*, *γ*, and *w*) to describe the creep behaviors for different gradations of CGS under different stresses, the nonlinear regression is needed. Since the Burgers model (Figure 6) is the Maxwell model connected to the Kelvin model in series, *E*_1_ and *η*_1_ are independent parameters, while *E*_2_ and *η*_2_ are coupling parameters. Therefore, the unified creep parameter relationship can be determined by nonlinear regression with high precision (*R*^2^ > 0.989).
(6)$$\begin{array}{ll} \mathrm{lg}(E_{1})= & 0.943+0.648\sigma -2.844\gamma -7.609w\;(R^{2}=0.989)\\ \mathrm{lg}(\eta_{1})= & 4.486+0.054\sigma -2.321\gamma -20.318w\;(R^{2}=0.992)\\ \mathrm{lg}(E_{2})= & -0.106+0.057\sigma +0.299\gamma +3.189w+0.472\mathrm{lg}(\eta_{2})\\ & -0.211{\mathrm{lg}}^{2}(\eta_{2})+0.377\mathrm{lg}(E_{2})\mathrm{lg}(\eta_{2})\;(R^{2}=0.998)\\ \mathrm{lg}(\eta_{2})= & 1.660-0.097\sigma -0.328\gamma -2.255w-0.156\mathrm{lg}(E_{2})\\ & -0.265{\mathrm{lg}}^{2}(E_{2})+0.451\mathrm{lg}(E_{2})\mathrm{lg}(\eta_{2})\;(R^{2}=0.999) \end{array}$$

Table 4 lists the nonlinearly regressed and tested values of Burgers creep parameters for comparison. They fit well, mostly with the error *E* < 10%, which proves the validity of Equation (6). Based on Burgers creep model and its parameter (Table 4), Figure 8 gives the tested and nonlinearly regressed creep curves (*ε_c_*–*t*) under different creep stresses for three gradations of CGS. It is seen that they are almost the same at the instantaneous deformation stage. In the attenuation deformation stage, their differences increase with time but tend to be unchanged in the stable deformation stage. In a word, they are approximately close to each other, which again proves the validity of Equation (6).

#### 2.2.3. Triaxial Fatigue Test and Analyze

(1)Fatigue compliance formulae

When the sine wave is adopted as the fatigue load:(7)$$\sigma =\sigma_{f}\sin\omega t$$ The strain response is
(8)$$\begin{array}{l} \varepsilon =\varepsilon_{f}\sin(\omega t-\delta )\\ \varepsilon_{f}=\sigma_{f}\left|J(\omega )\right| \end{array}$$
where *σ_f_* and *ε_f_* are the peak stress and strain of fatigue, *δ* is the phase angle and |*J*(*ω*)| is the module of fatigue compliance.

Equations (7) and (8) can be rewritten as
(9)$$\frac{\sigma }{\sigma_{f}}=\sin\omega t$$
(10)$$\frac{\varepsilon }{\varepsilon_{f}}=\sin\omega t\cos\delta -\sqrt{1-\sin^{2}\omega t}\sin\delta$$

Substituting Equation (9) into Equation (10) yields the strain–stress relationship, i.e., the elliptic Equation:(11)$$\frac{\sigma^{2}}{\sigma_{f}^{2}}+\frac{\varepsilon^{2}}{\varepsilon_{f}^{2}}-\frac{2\cos\delta }{\sigma_{f}\varepsilon_{f}}\sigma \varepsilon -\sin^{2}\delta =0$$

Since this elliptic Equation is obtained by fitting the data of a one-cycle fatigue test, the phase angle *δ* can be determined by Equation (11).

Therefore, the real parts *J_r_*(*ω*) and imaginary parts *J_i_*(*ω*) of the fatigue compliance can be calculated by Equation (8) and the relationship of the phase angle *δ* and the fatigue compliance [23] are as follows:(12)$$\begin{array}{l} \tan\delta =\frac{J_{i}(\omega )}{J_{r}(\omega )}\\ \left|J(\omega )\right|=\sqrt{J_{r}{\left(\omega \right)}^{2}+J_{i}{\left(\omega \right)}^{2}} \end{array}$$

(2)Experimental determination of fatigue compliance

Dynamic loading adopts the stress-controlled cyclic loading method, applying semi-sine wave axial cyclic stress, and the fatigue load was applied on the specimen after the balance of *σ*_1_ = *σ*_3_, and recorded 10 points of the stress and strain in one cycle by the test system. Figure 9 shows some specimens after the fatigue test of different gradations of CGS.

Figure 10 and Figure 11 show the tested and fitted hysteretic curves of the S1 specimen (at different frequency *f* for the same average stress *σ_a_*) and the S2 specimen (under different *σ**_a_* for the same *f*) for a one-cycle load for example. Table 5 lists the areas of the hysteretic circles (*A* represents the energy dissipation in one cycle, Equation (13) [27]) obtained by Matlab with its own written code. It can be seen that *A* slightly decreased with the increase in *f*, that is, the frequency effect is much smaller, while *A* was almost multiplied as it increased with the multiplied increasing of *σ_a_* for the same gradation CGS. Additionally, *A* increased with the decrease in pebble content (*γ*) for different gradations of CGS, because a greater *γ* means that the CGS specimen had weaker rheological properties; therefore, the smaller *γ* represents the stronger rheological properties which could dissipate more energy in one cycle. The fatigue compliance (*J_r_*(*ω*), *J_i_*(*ω*), *J*(*ω*) and *δ*) can be calculated by Equations (7)–(13).
(13)$$\begin{array}{ll} A & =\pi \sigma_{f}\varepsilon_{f}\sin\delta \\ & =\pi \sigma_{f}^{2}J_{i}(\omega ) \end{array}$$

#### 2.2.4. Determination of Creep–Fatigue Interaction Parameters

The creep–fatigue interaction parameters *k* and *g* can be determined by Equation (4) when the real part *J_r_*(*t*) and imaginary part *J_i_*(*t*) of the creep compliance *J*(*t*) is calculated by substituting Equation (5) with Table 4 into Equation (2), and the real part *J_r_*(*ω*) and imaginary part *J_i_*(*ω*) of the fatigue compliance *J_i_*(*ω*) is calculated by Equation (12). Table 6, Table 7 and Table 8 list the calculation results of creep and fatigue compliance for different gradations of CGS under different stress levels (*σ_a_*).

It can be seen that with the increasing frequency *f* that the real part *J_r_*(*t*) of creep compliance (representing the energy storage compliance) is unchanged while the imaginary part *J_i_*(*t*) (representing the energy dissipation compliance) is deceased, which is the same as analytic results [28]. With the decreasing pebble content *γ, J_r_*(*t*) is decreased but *J_i_*(*t*) is increased, since a smaller value of *γ* means a greater clay content of the CGS specimen and thus stronger rheological properties with more energy dissipation in one cycle.

Similarly, with the increase in frequency *f*, the real part *J_r_*(*ω*) of fatigue compliance is unchanged while the imaginary part *J_i_*(*ω*) is deceased. With the decrease in *γ*, *J_r_*(*ω*) is decreased but *J_i_*(*ω*) is increased.

Obviously, the creep–fatigue interaction parameter *k* (corresponding to *J_r_*(*t*) and *J_r_*(*ω*)) is also unchanged while *g* (corresponding to *J_i_*(*t*) and *J_i_*(*ω*)) is increased with the increase in frequency *f*. With the decrease in *γ, k* changes non-monotonically while *g* is increased.

## 3. Test Verification of the Creep–Fatigue Interaction Model

### 3.1. Prediction Method and Test Scheme

Take the creep–fatigue interaction deformation of CGS as an example (the fatigue–creep interaction deformation is vice versa): the strain–time (*ε*–*t*) curve can be calculated by the following steps: (1) calculate *J_r_*(*t*) and *J_i_*(*t*) by Equation (2) to obtain the *ε_c_*–*t* curve; (2) calculate *J_r_*(*ω*) and *J_i_*(*ω*) by Equation (12) to obtain the *ε_f_*–*t* curve; (3) according to the creep–fatigue interaction factors *k* and *g* in Table 6, Table 7 and Table 8 to calculate the creep–fatigue interaction *ε*–*t* curve.

To verify the validity of the calculated *ε*–*t* curve based on the new creep–fatigue interaction model, triaxial creep–fatigue interaction tests, triaxial fatigue–creep interaction tests and the triaxial multi-interaction of three gradations of CGS specimens (Table 1) were conducted by triaxial test system (Figure 5). Table 9 lists the loading conditions of creep–fatigue interaction tests for different gradations of CGS under the same confining pressure (0.2 MPa).

### 3.2. Results and Analysis

#### 3.2.1. Triaxial Creep–Fatigue Interaction Deformation

Figure 12 shows the predicted and tested curves of creep–fatigue (C-F) interaction deformation for S1 specimens under different stress levels. It is seen that all creep curves (the first part of C-F interaction) and fatigue curves (the second part of C-F interaction) have three stages: (1) instantaneous deformation stage; (2) attenuation deformation stage; and (3) stable deformation stage. The predicted creep and fatigue curves are close to the tested curves, which can prove the validity of the new creep–fatigue interaction model. The difference between the predicted and tested fatigue curves might be caused by the impact effects of fatigue loads and the discreteness of geotechnical materials.

#### 3.2.2. Triaxial Fatigue–Creep Interaction Deformation

Figure 13 shows the predicted and tested *ε*–*t* curves of fatigue–creep interaction for S2 specimens. It can be seen that all of the creep curves and all of the fatigue creep curves also have three deformation stages. In the fatigue stage, there is no situation in which the tested strain curve is much larger than the predicted strain curve compared with Figure 11, because the creep complex compliance is calculated based on the fatigue complex compliance and the predicted and tested curves are close to each other. In the creep stage, the predicted and tested curves are in good agreement. This again verifies the validity of the new creep–fatigue interaction model.

#### 3.2.3. Triaxial Creep–Fatigue Multi-Interaction Deformation

Figure 14 shows the predicted and tested curves of creep–fatigue multi-interaction for S3 specimen. It is seen that the predicted curves are in good agreement with the tested curves for two creep stages, while the predicted curves are larger than the tested curves for two fatigue stages. The reason is the dynamic load make a greater deformation for the same average stress. For the two fatigue stages, the strain produced from 120 min to 240 min is approximately 0.8%, while the strain produced from 360 min to 480 min is approximately 0.6%; the strain of the second fatigue stage is smaller than that of the first fatigue stage, because the rough pores of the CGS have been irreversibly reduced in the first fatigue stage, and after the first fatigue stage, the internal particles of the CGS will form a more stable force chain; hence, the strain generated in the second fatigue stage is less than the first fatigue stage.

## 4. Conclusions

(1)New transformation parameters (*k* and *g*) are proposed to establish a new creep–fatigue interaction model based on both the rheologic mechanics and the interactive relationship between creep and fatigue complex compliance, in order to predict the creep–fatigue interaction deformation of CGS.(2)The creep–fatigue interaction factor *k* is almost unchanged with the frequency (*f*) and changed non-monotonic with the pebble content (*γ*). The creep–fatigue interaction factor *g* is increased with both *f* and *γ*. The creep strain and fatigue strain of the creep–fatigue and fatigue–creep interaction are increased with the increase in average stress (*σ_a_*)(3)For the different interaction orders (creep–fatigue interaction or fatigue–creep interaction), the fatigue deformation is always larger than the creep deformation under the same stress level because of the dynamic effect.(4)For the creep–fatigue multi-interaction, the second creep and fatigue deformation is always smaller than the first creep and fatigue deformation because of the compaction effect.(5)The new creep–fatigue interaction model is proved valid by the good agreement between the predicted results and test results of the triaxial creep–fatigue interaction. It can be further developed for predicting the creep–fatigue interaction of multi-layered coarse-grained soil.

## Figures and Tables

**Figure 1 materials-15-03904-f001:**
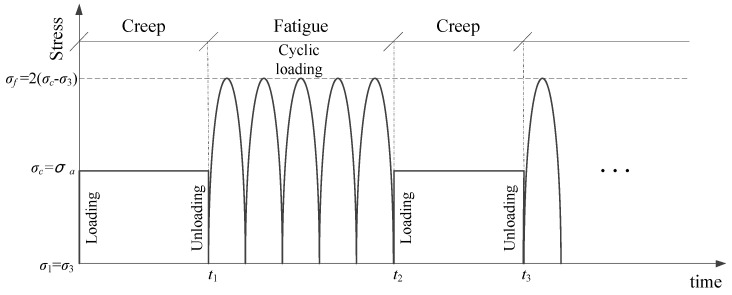
The process of creep–fatigue interaction.

**Figure 2 materials-15-03904-f002:**
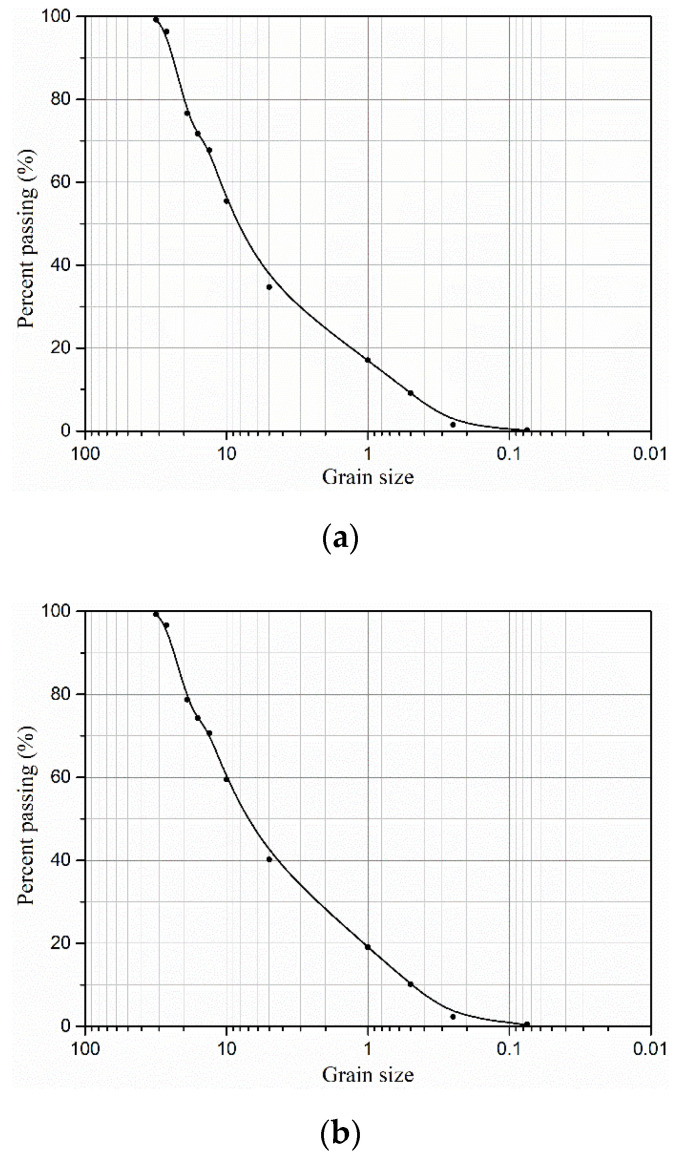
Grain–size distributions curves for three different gradations of CGS: (**a**) S1; (**b**) S2; and (**c**) S3.

**Figure 3 materials-15-03904-f003:**
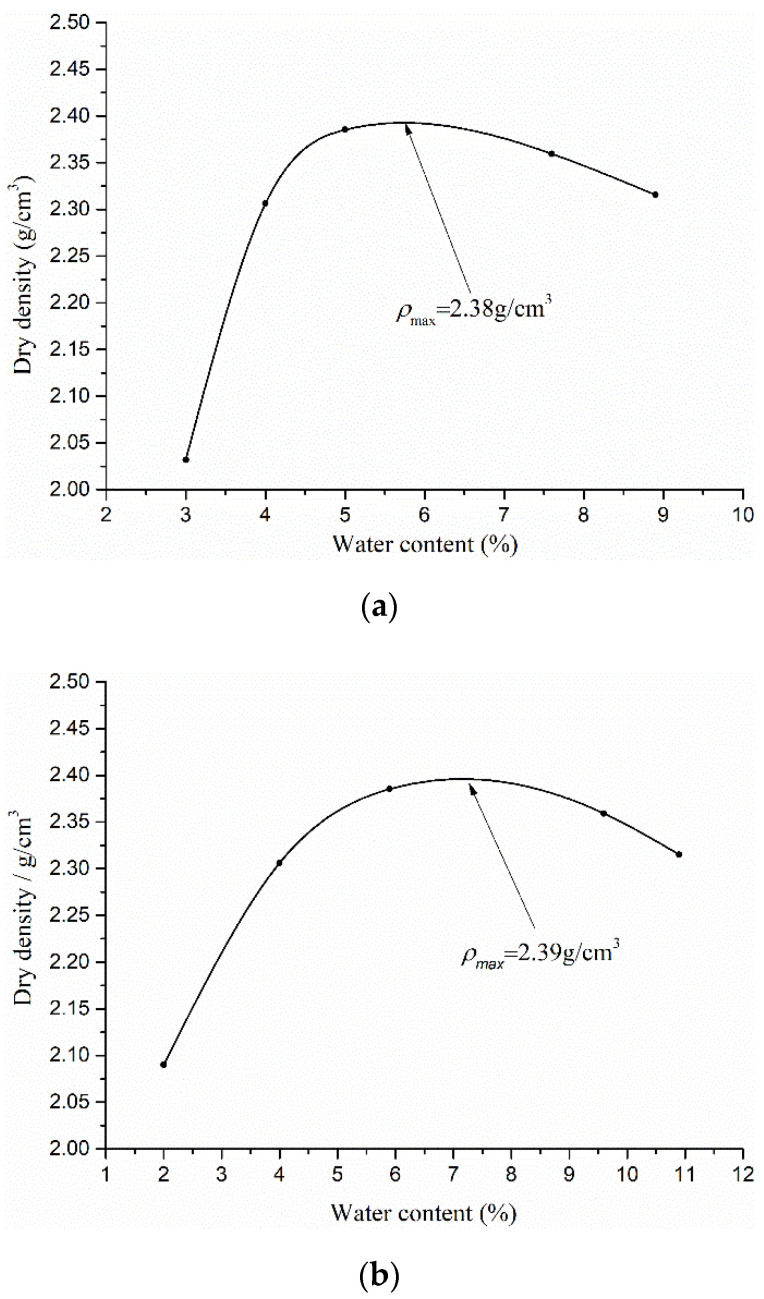
Water content–dry density curves for three different gradations of CGS: (**a**) S1; (**b**) S2; and (**c**) S3.

**Figure 4 materials-15-03904-f004:**
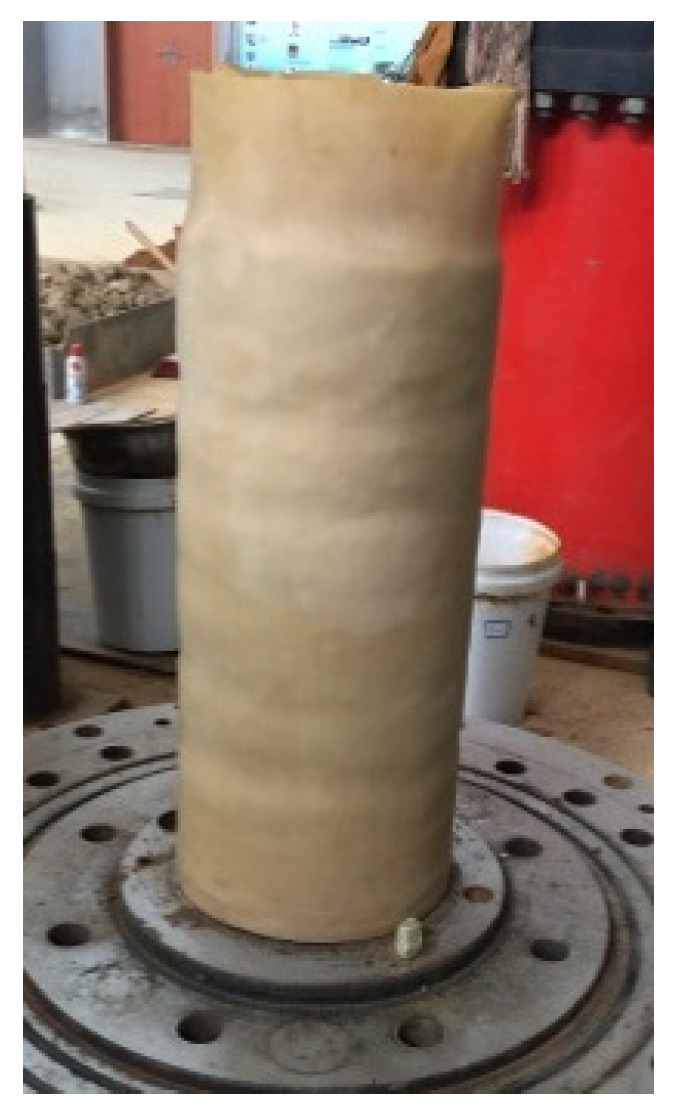
Cylindric specimen of CGS.

**Figure 5 materials-15-03904-f005:**
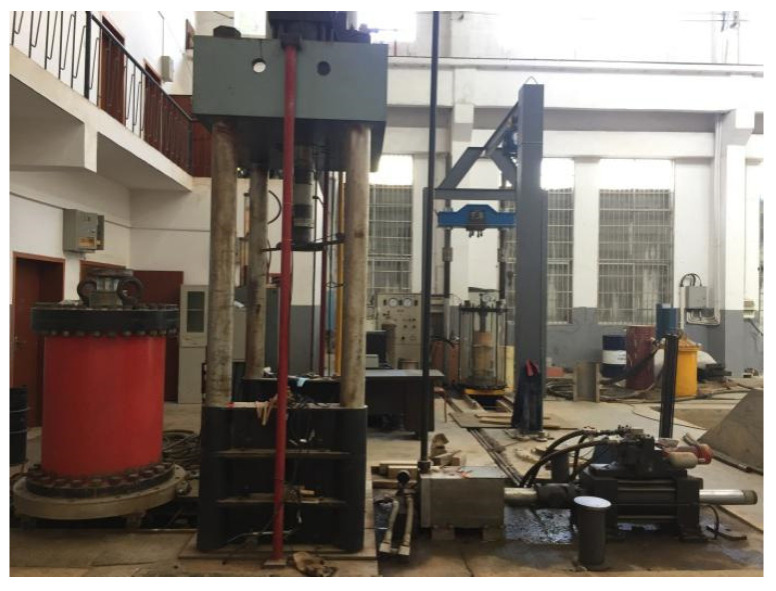
TAJ-2000 triaxial testing system.

**Figure 6 materials-15-03904-f006:**
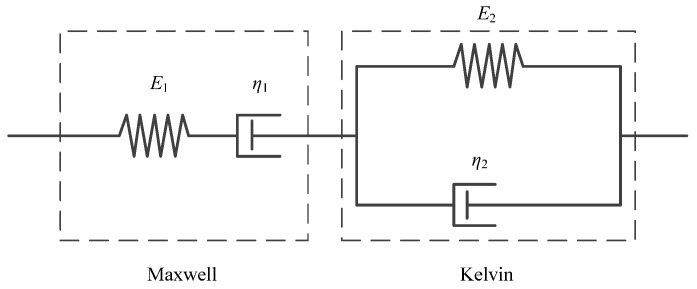
Burgers model.

**Figure 7 materials-15-03904-f007:**
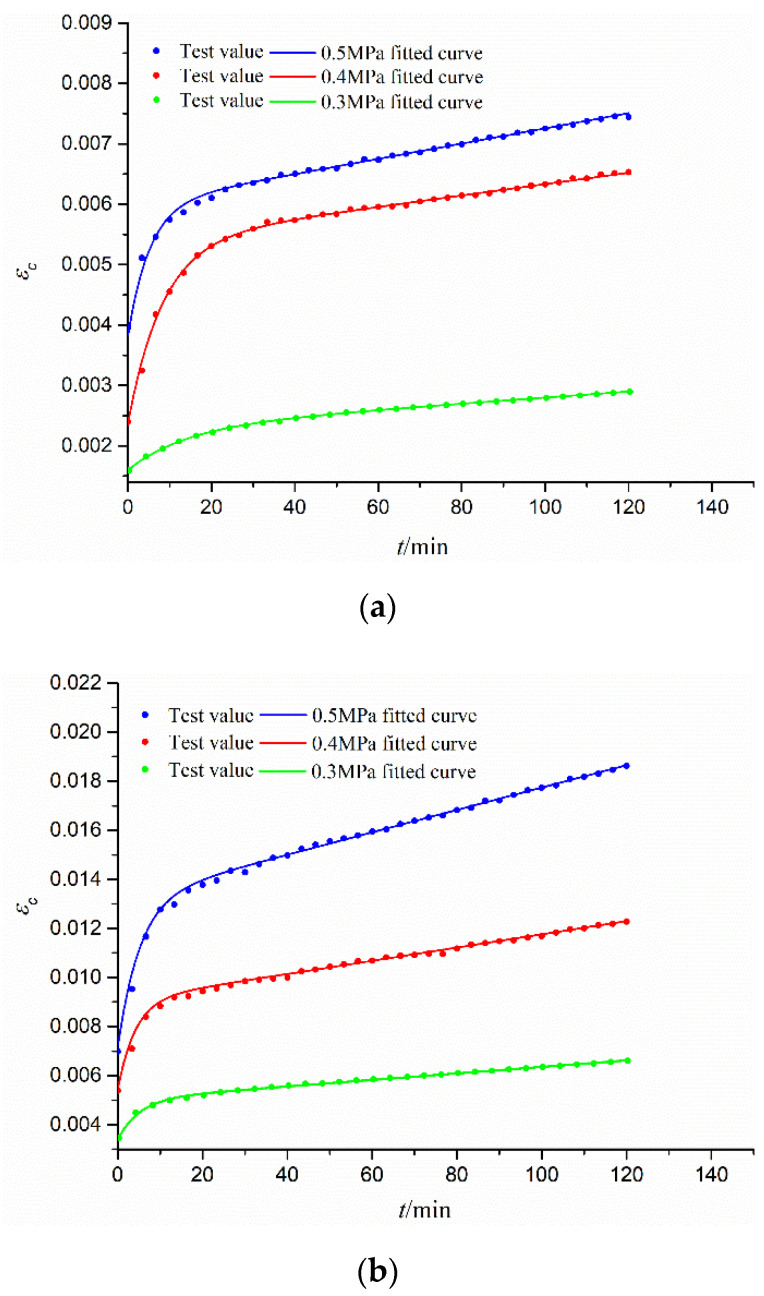
Creep curves for different gradations of CGS specimens: (**a**) S1; (**b**) S2; and (**c**) S3.

**Figure 8 materials-15-03904-f008:**
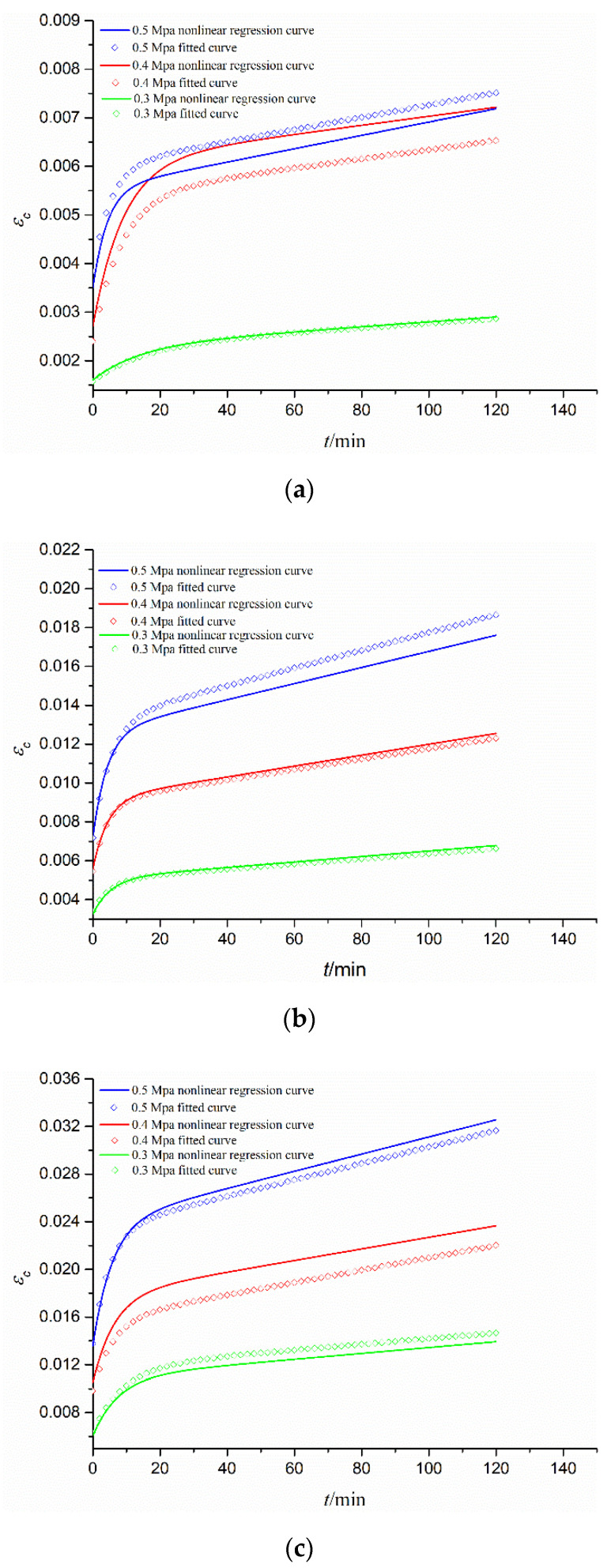
Fitted and nonlinearly regressed creep curves under different stresses for different gradations of CGS: (**a**) S1; (**b**) S2; and (**c**) S3.

**Figure 9 materials-15-03904-f009:**
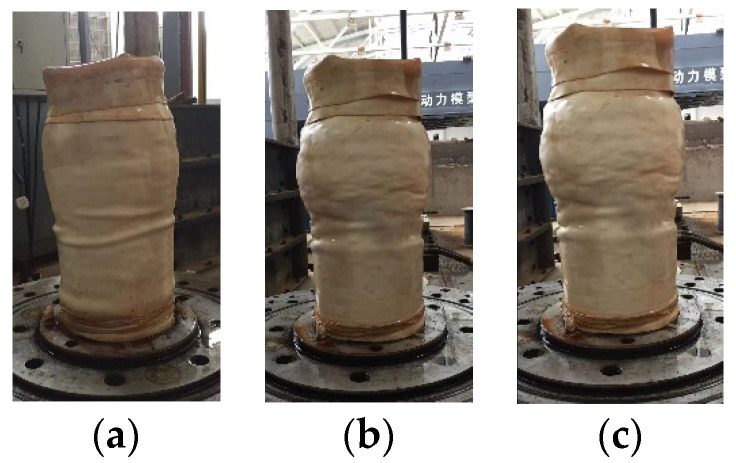
Deformation of CGS after fatigue tests: (**a**) S1; (**b**) S2; and (**c**) S3.

**Figure 10 materials-15-03904-f010:**
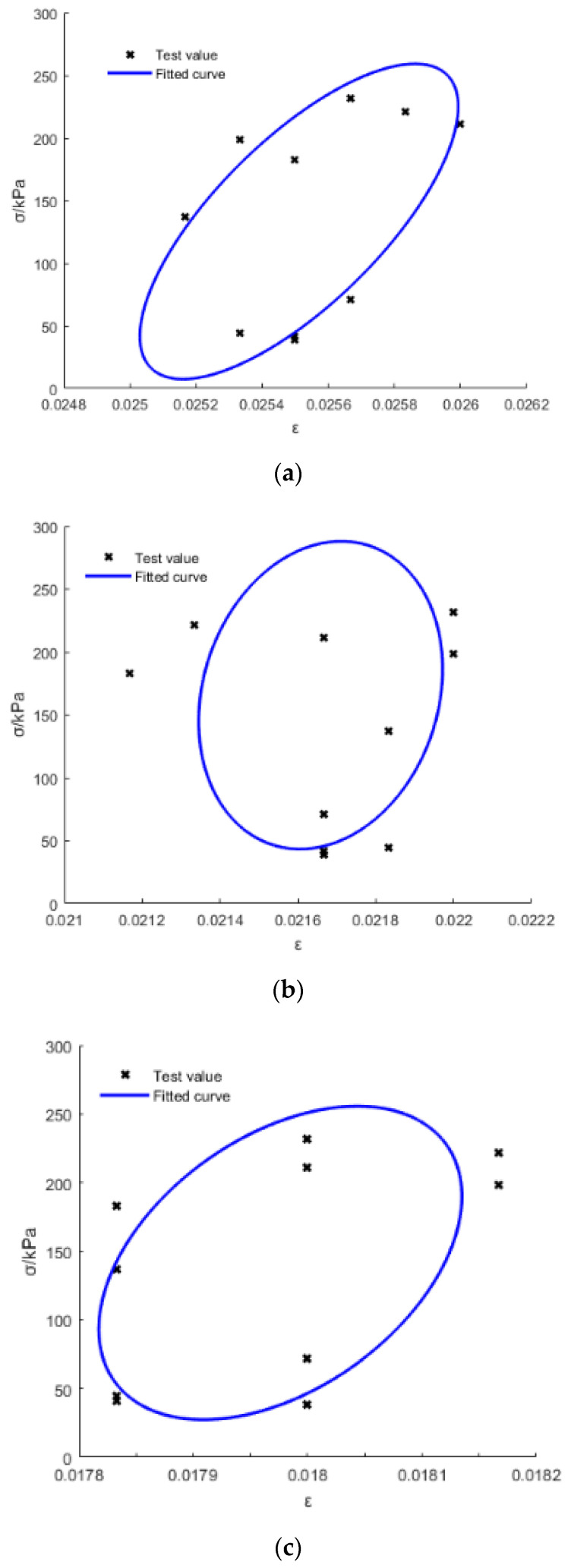
Tested and fitted hysteretic curves of the S1 specimen under different frequencies: (**a**) 1 Hz; (**b**) 2 Hz; and (**c**) 3 Hz.

**Figure 11 materials-15-03904-f011:**
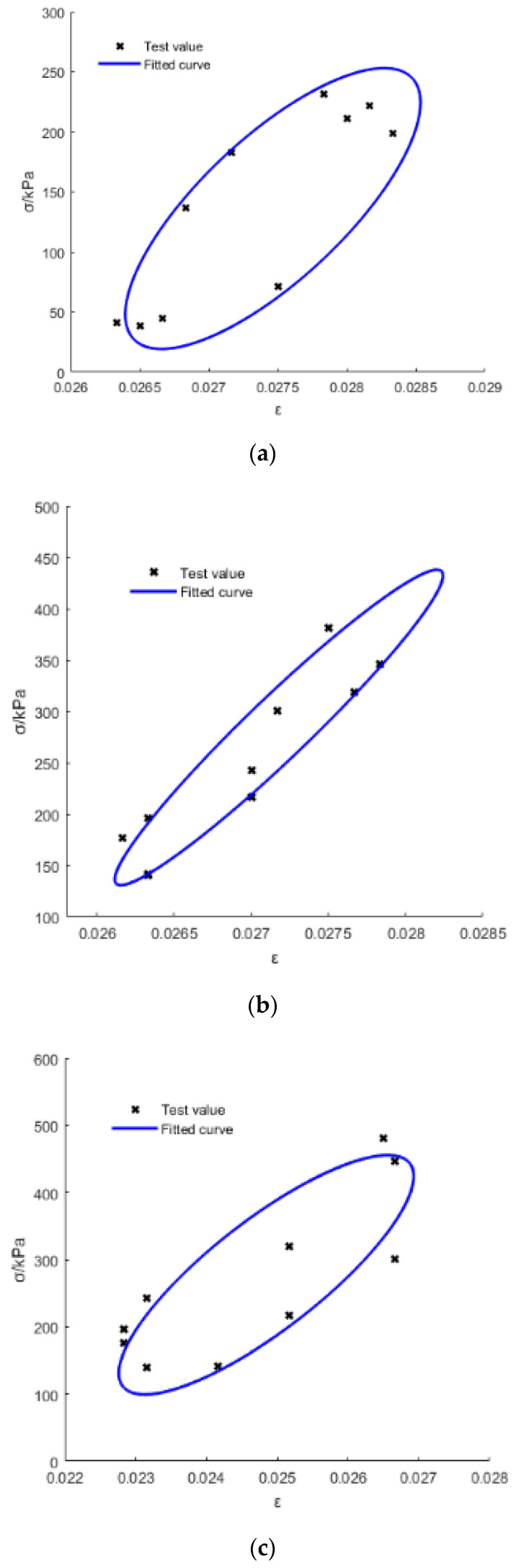
Tested and fitted hysteretic curves of the S2 specimen under different average stress: (**a**) 0.1 Mpa; (**b**) 0.2 Mpa; and (**c**) 0.3 Mpa.

**Figure 12 materials-15-03904-f012:**
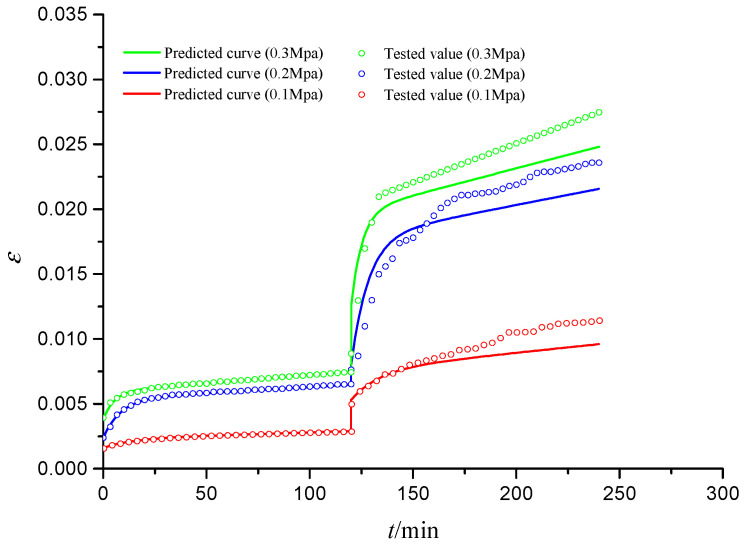
Predicted and tested curves of creep–fatigue interaction deformation (S1).

**Figure 13 materials-15-03904-f013:**
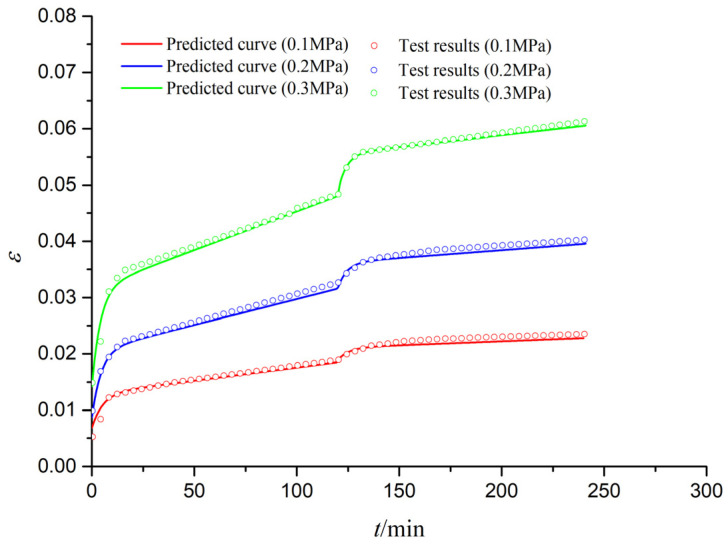
Predicted and tested curves of the fatigue–creep interaction deformation (S2).

**Figure 14 materials-15-03904-f014:**
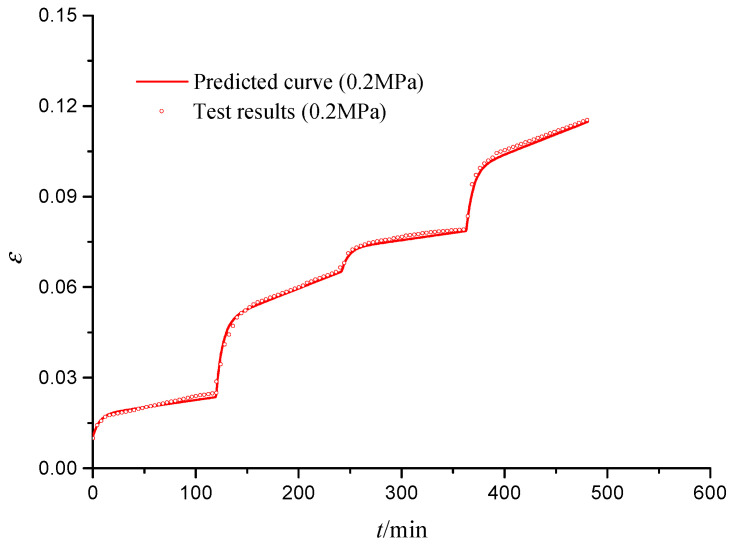
Predicted and tested curves of fatigue–creep multi-interaction deformation (S3).

**Table 1 materials-15-03904-t001:** Three different mass ratios of CGS.

No.	Pebble (*γ*)	Sand	Clay
S1	1	1	0.30
S2	1	1	0.75
S3	1	1	1.33

**Table 2 materials-15-03904-t002:** Loading conditions for the triaxial compressive creep and fatigue tests (*σ*_3_ = 0.2 MPa).

No.	Creep		Fatigue			
	*σ_c_*/MPa	Holding Time *t*/min	Peak Stress *σ_f_*/MPa	Frequency *f*/Hz	Cycle Number num/N	Average Stress *σ_a_*/MPa
S1/S2/S3	0.3	120	0.4	1	7200	0.1
				2	14,400	
				3	21,600	
	0.4		0.6	1	7200	0.2
				2	14,400	
				3	21,600	
	0.5		0.8	1	7200	0.3
				2	14,400	
				3	21,600	

**Table 3 materials-15-03904-t003:** Burgers model parameters for different gradations of CGS.

No.	*σ_c_* − *σ*_3_/MPa	*E*_1_/MPa	*E*_2_/MPa	*η*_1_/MPa·min	*η*_2_/MPa·min	*R* ^2^
S1	0.1	62.51	142.95	19,958.67	1974.63	0.997
	0.2	83.33	66.67	21,276.59	555.56	0.998
	0.3	78.13	138.88	23,809.52	730.99	0.996
S2	0.1	29.36	61.22	7544.07	312.03	0.998
	0.2	36.49	55.56	7407.40	229.56	0.996
	0.3	41.67	50.00	6578.94	257.73	0.996
S3	0.1	15.74	18.33	4196.82	163.64	0.998
	0.2	20.28	33.67	3846.15	199.23	0.997
	0.3	21.73	31.25	4347.82	157.82	0.995

**Table 4 materials-15-03904-t004:** Regressed and tested values of Burgers creep parameters.

No.		S1			S2			S3		
*σ_c_*/MPa		0.3	0.4	0.5	0.3	0.4	0.5	0.3	0.4	0.5
*E*_1_/ MPa	Tested value	62.51	83.33	78.13	29.36	36.49	41.67	15.74	20.28	21.73
	Calculated value	63.65	73.89	85.78	30.47	35.37	41.07	16.38	19.02	22.08
	Error	1.82%	11.32%	9.79%	3.78%	3.06%	0.05%	4.06%	6.21%	1.61%
*E*_2_/ MPa	Tested value	142.95	66.67	138.88	61.22	55.56	50.00	18.33	33.67	31.25
	Calculated value	138.02	58.88	146.98	54.95	56.23	57.07	20.41	27.28	28.96
	Error	3.44%	11.68%	5.81%	10.24%	1.21%	14.14%	11.34%	18.90%	7.32%
*η*_1_/ MPa·min	Tested value	19,958.67	21,276.59	23,809.52	7544.07	7407.40	6578.94	4196.82	3846.15	4347.82
	Calculated value	21,300.86	21,567.37	21,837.21	7057.92	7146.23	7235.64	4064.62	4115.48	4166.97
	Error	6.72%	1.36%	8.28%	6.44%	3.52%	9.98%	3.15%	7.00%	4.15%
*η*_2_/ MPa·min	Tested value	1974.63	555.56	730.99	312.03	229.56	257.73	163.64	199.23	157.82
	Calculated value	2092.09	531.02	630.95	293.26	251.18	239.88	158.48	176.42	165.95
	Error	5.94%	4.41%	13.68%	6.01%	9.41%	6.92%	3.15%	11.44%	5.15%

**Table 5 materials-15-03904-t005:** Area of hysteretic circle (*A*) for different CGS specimens (unit: N·m/m^3^).

No.	*σ_a_*/MPa	0.1			0.2			0.3		
	*f*/Hz	1	2	3	1	2	3	1	2	3
S1		97.26	96.45	96.02	201.54	198.61	193.21	298.45	284.25	280.62
S2		187.36	184.56	183.27	387.58	385.12	376.31	572.54	568.14	557.10
S3		243.95	236.57	230.47	497.66	485.74	472.96	740.57	724.65	719.45

**Table 6 materials-15-03904-t006:** Calculation results of creep–fatigue interaction parameters *k* and ***g*** for different gradations of CGS (*σ_a_* = 0.1 MPa).

No.	*γ*	*f*/Hz	Creep		Fatigue		Matrix(*M*)
			*J_r_*(*t*)/Mpa^−1^	*J_i_*(*t*)/Mpa^−1^	*J_r_*(*ω*)/Mpa^−1^	*J_i_*(*ω*)/Mpa^−1^	
S1	0.434	1	0.0157	0.000083	0.0531	0.001535	$\left(\begin{array}{cc} 3.38 & 0\\ 0 & 18.49 \end{array}\right)$
		2	0.0157	0.000041	0.0530	0.001314	$\left(\begin{array}{cc} 3.38 & 0\\ 0 & 32.04 \end{array}\right)$
		3	0.0157	0.000027	0.0532	0.001054	$\left(\begin{array}{cc} 3.39 & 0\\ 0 & 39.03 \end{array}\right)$
S2	0.363	1	0.0328	0.000565	0.1143	0.006992	$\left(\begin{array}{cc} 3.48 & 0\\ 0 & 12.37 \end{array}\right)$
		2	0.0328	0.000282	0.1145	0.005854	$\left(\begin{array}{cc} 3.49 & 0\\ 0 & 20.75 \end{array}\right)$
		3	0.0328	0.000188	0.1142	0.004101	$\left(\begin{array}{cc} 3.48 & 0\\ 0 & 21.81 \end{array}\right)$
S3	0.272	1	0.0611	0.001043	0.1754	0.013388	$\left(\begin{array}{cc} 2.87 & 0\\ 0 & 12.83 \end{array}\right)$
		2	0.0611	0.000521	0.1756	0.009633	$\left(\begin{array}{cc} 2.87 & 0\\ 0 & 18.48 \end{array}\right)$
		3	0.0611	0.000347	0.1758	0.005702	$\left(\begin{array}{cc} 2.88 & 0\\ 0 & 16.43 \end{array}\right)$

**Table 7 materials-15-03904-t007:** Calculation results of creep–fatigue interaction parameters *k* and *g* for different gradations of CGS (*σ_a_* = 0.2 MPa).

No.	*γ*	*f*/Hz	Creep		Fatigue		Matrix(*M*)
			*J_r_*(*t*)/Mpa^−1^	*J_i_*(*t*)/Mpa^−1^	*J_r_*(*ω*)/Mpa^−1^	*J_i_*(*ω*)/Mpa^−1^	
S1	0.434	1	0.0135	0.000307	0.0512	0.002203	$\left(\begin{array}{cc} 3.79 & 0\\ 0 & 7.17 \end{array}\right)$
		2	0.0135	0.000153	0.0513	0.002015	$\left(\begin{array}{cc} 3.79 & 0\\ 0 & 13.16 \end{array}\right)$
		3	0.0135	0.000102	0.0512	0.001674	$\left(\begin{array}{cc} 3.79 & 0\\ 0 & 16.41 \end{array}\right)$
S2	0.363	1	0.0282	0.000655	0.1282	0.011689	$\left(\begin{array}{cc} 4.54 & 0\\ 0 & 17.84 \end{array}\right)$
		2	0.0282	0.000328	0.1284	0.010494	$\left(\begin{array}{cc} 4.55 & 0\\ 0 & 31.99 \end{array}\right)$
		3	0.0282	0.000218	0.1283	0.008829	$\left(\begin{array}{cc} 4.55 & 0\\ 0 & 40.50 \end{array}\right)$
S3	0.272	1	0.0525	0.000940	0.2155	0.027809	$\left(\begin{array}{cc} 4.10 & 0\\ 0 & 29.58 \end{array}\right)$
		2	0.0525	0.000471	0.2153	0.022021	$\left(\begin{array}{cc} 4.09 & 0\\ 0 & 46.75 \end{array}\right)$
		3	0.0525	0.000313	0.2154	0.018344	$\left(\begin{array}{cc} 4.10 & 0\\ 0 & 58.61 \end{array}\right)$

**Table 8 materials-15-03904-t008:** Calculation results of creep–fatigue interaction parameters *k* and *g* for different gradations of CGS (*σ_a_* = 0.3 MPa).

No.	*γ*	*f*/Hz	Creep		Fatigue		Matrix(*M*)
			*J_r_*(*t*)/Mpa^−1^	*J_i_*(*t*)/Mpa^−1^	*J_r_*(*ω*)/Mpa^−1^	*J_i_*(*ω*)/Mpa^−1^	
S1	0.434	1	0.0116	0.000259	0.0857	0.006082	$\left(\begin{array}{cc} 7.38 & 0\\ 0 & 23.48 \end{array}\right)$
		2	0.0116	0.000129	0.0858	0.005106	$\left(\begin{array}{cc} 7.38 & 0\\ 0 & 39.58 \end{array}\right)$
		3	0.0116	0.000086	0.0855	0.004116	$\left(\begin{array}{cc} 7.37 & 0\\ 0 & 47.86 \end{array}\right)$
S2	0.363	1	0.0243	0.000684	0.1433	0.017303	$\left(\begin{array}{cc} 5.89 & 0\\ 0 & 25.29 \end{array}\right)$
		2	0.0243	0.000342	0.1435	0.015152	$\left(\begin{array}{cc} 5.90 & 0\\ 0 & 44.30 \end{array}\right)$
		3	0.0243	0.000228	0.1436	0.012881	$\left(\begin{array}{cc} 5.90 & 0\\ 0 & 56.49 \end{array}\right)$
S3	0.272	1	0.0453	0.000997	0.2874	0.037071	$\left(\begin{array}{cc} 6.34 & 0\\ 0 & 37.18 \end{array}\right)$
		2	0.0453	0.000498	0.2872	0.035391	$\left(\begin{array}{cc} 6.34 & 0\\ 0 & 71.06 \end{array}\right)$
		3	0.0453	0.000332	0.2871	0.031228	$\left(\begin{array}{cc} 6.33 & 0\\ 0 & 94.06 \end{array}\right)$

**Table 9 materials-15-03904-t009:** Loading conditions for triaxial creep–fatigue tests.

No.	Loading Order (C-F Interaction)	Creep		Fatigue		
		*σ_c_*/Mpa	Holding Time (*t*)/min	*σ_a_*/Mpa	*f*/Hz	Cycle Number/N
S1	C-F	0.1/0.2/0.3	120	0.1/0.2/0.3	1	7200
S2	F-C	0.1/0.2/0.3	120	0.1/0.2/0.3	1	7200
S3	C-F-C-F	0.2	120	0.2	1	14,400

## Data Availability

Not applicable.

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
