# Peer review of "A New Creep–Fatigue Interaction Model for Predicting Deformation of Coarse-Grained Soil"

_materials, 2022, doi:10.3390/ma15113904_

Round 1
Reviewer 1 Report
This work proposed to establish the new creep-fatigue interaction model based on rheology mechanics and the relationship between creep and fatigue complex compliance. The procedures of experiment and establishing new creep-fatigue interaction model are well explained to understand. The fitted parameters before Fig. 7 seems to be good to predict/explain experimental results. However, nonlinearly-regressed creep curves under different stress for different gradations CGS (Fig. 8), the model did not fit well especially for S1 and S3. Also, the predicted creep-fatigue interaction deformation in Fig. 11 does not match well with experimental results. I’m not sure that it might be caused by the impact effects of fatigue loads as authors described in manuscript.
Reviewer 2 Report
The article presents the current topic of A New Creep-fatigue Interaction Model for Predicting Deformation of Coarse-grained Soil. While the work presents an interesting study, it requires a significant revision before it is recommended for publication.
1. Kindly add how the fatigue tests were conducted (with experimental pictures).
2. In Fig. 2 Please add the high quality images.
3. In Table 3, Kindly add the Units in the second row.
4. In Fig. 9, 10 Please improve the quality of the images.
5. Improve the Conclusion section based on the experimental and predicted results.
Round 2
Reviewer 1 Report
I think the error (~18%) seems to be quite high, but authors mentioned that this level is acceptable in field. As further improvement will be difficult, I propose to accept this manuscript.
Reviewer 2 Report
The authors mitigated all reviewer's queries and highlighted them in the revised manuscript. Therefore, it might be acceptable for publication in the present form.